# Patient perspectives on the HIV continuum of care in London: a qualitative study of people diagnosed between 1986 and 2014

Jane Bruton,[1] Tanvi Rai,[1] Sophie Day,[1,2] Helen Ward[1]

## ABSTRACT

**Objectives** To describe the experiences of the HIV treatment cascade of diagnosis, engagement with care and initiation of treatment from the perspective of patients; we explored whether this differed according to the year of their diagnosis, for example, whether they had experienced HIV care in the pretreatment era.

**Design** Qualitative interview study with framework analysis.

**Setting** Two large HIV adult outpatient clinics in central London.

**Participants** 52 HIV-positive individuals, 41 men, 11 women, purposively sampled to include people who had been diagnosed at different stages in the history of the epidemic classified as four 'generations': pre-1996 (preantiretroviral therapy (ART)), 1997–2005 (complex ARTs), 2006–2012 (simpler ARTs) and 2013 onwards (recent diagnoses).

**Results** Some important differences were identified; for earlier generations, the visible illness and deaths from AIDS made it harder to engage with care following diagnosis. Subsequent decisions about starting treatment were deeply influenced by the fear of severe side effects from early ART. However, despite improvements in ART and life expectancy over the epidemic, we found a striking similarity across participants' accounts of the key stages of the care continuum, regardless of when they were diagnosed. Diagnosis was a major traumatic life event for almost everyone. Fear of testing positive or having low self-perceived risk affected the timing of testing and diagnosis. Engaging with care was facilitated by a flexible approach from services/clinicians. Initiating treatment was a major life decision.

**Conclusion** We found patients' experiences are influenced by when they were diagnosed, with earliest cohorts facing substantial challenges. However, being diagnosed with HIV and starting treatment continue to be significant life-altering events even in the era of effective, simple treatments. Despite the advances of biomedical treatment, services should continue to recognise the needs of patients for whom the diagnosis and treatment remain significant challenges.

### Strengths and limitations of this study

► The large number of participants at two different clinics and the inclusion of a range of people with HIV, broadly similar to the clinic cohorts, across four HIV 'generations'.

► The interview format allowed us to explore factors important to participants rather than asking about predefined concerns.

► An imbalance between generations, with larger numbers of women in the earliest two generations and only one woman in the generation recently diagnosed. This means that some of our conclusions about generations may also reflect gendered differences.

► Our recruitment methods meant we were not able to explore the experiences of those who remain outside care.

► Limitation of the single-interview format means people were having to recall past experiences, some of which were three decades ago. This may have introduced recall bias with subsequent experience colouring earlier reports.

[1]Department of Infectious Disease Epidemiology, Imperial College London, London, UK
[2]Department of Anthropology, Goldsmiths University of London, London, UK

**Correspondence to**
Mrs Jane Bruton;
jbruton@ic.ac.uk

## INTRODUCTION

The HIV continuum of care provides step-wise estimates for the stages of engagement in care for people living with HIV (PLHIV).[1] The Joint United Nations Programme on HIV and AIDS's goal to end the AIDS epidemic by 2020 stipulates a target of 90% of all people with HIV be diagnosed, of whom 90% are on antiretroviral therapy (ART), of whom 90% (73% of PLHIV) are virally suppressed.[2] The UK has one of the best outcomes for HIV treatment and care in the world with an estimated 78% of PLHIV having an undetectable viral load.[3 4] However, the statistics do not and cannot tell us about the patient experience of passing through these stages of care whether it be good or bad. The continuum essentially measures the success of programmes from a provider rather than a patient perspective.[5]

Flowers et al[4] argue that there is a tension between the certainty and confidence of a linear HIV pathway, associated with ideas of clinical efficacy, and patient experiences of

diagnosis and prognosis, which can be full of uncertainty. With the evolution of modern ARTs, we have witnessed the transformation of HIV from an acute life-threatening infection to a treatable chronic condition and a concurrent evolving of the care continuum. However, it is unclear whether this change is reflected in patients' own experiences of passing through each of the stages of care. For example, has the moment of diagnosis become any less traumatic, and have decisions about starting treatment become simpler for patients? Analysis of patient narratives, historically and currently, may help to highlight significant factors for patients in the care continuum. This is particularly important in a climate of National Health Service restructuring in the UK, which has led to reductions in non-clinical services and streamlining of care. To this end, we explored the patient perspectives on the care continuum, and we hypothesised that patients' experiences of, and engagement with, care would differ according to what point in the epidemic they were diagnosed.

## METHODS

We undertook a qualitative study of people attending two public HIV clinics in London that have provided care since the start of the epidemic; they were also chosen for their large size and diversity, in terms of demographics. They are both specialist HIV clinics linked to sexual health (genitourinary medicine) services. Care is provided by physician-led multidisciplinary teams where patients have a named consultant. HIV care in the UK is free and open access, allowing patients to register at their clinic of choice. We used a purposive sampling method to recruit patients with a range of experiences. To reflect the evolution of ART, we identified four 'generations' according to time of diagnosis: pre-1996 (pre-ART), 1997–2005 (complex ARTs), 2006–2012 (simpler ARTs) and 2013 onwards (recent diagnoses). Within each generation, we aimed to include people with a range of characteristics, including, gender, exposure, age and ethnicity. Participants were recruited opportunistically by researchers attending clinical services and through fliers and digital advertising in clinical areas. Recruitment was periodically checked against the recruitment matrix and under-represented groups/strata targeted.

Patients were provided with information and gave written consent. Interviews took place in private rooms in or near the clinics or at the patient's home; they were recorded and transcribed, and interviews lasted between 60 and 90 minutes. The interviews were semistructured and carried out by one male (CH) and three female researchers (JB, TR and JR), three of whom had clinical backgrounds. The interviews were based on a topic guide (see online supplementary file) informed by a focus group of PLHIV who assisted in designing the research. We invited participants to recall their initial diagnosis and describe key points in their HIV journey including testing, disclosure, support, engaging with care, starting treatment, medication adherence, work and social life. Field notes were written after the interviews.

Transcripts were uploaded to NVivo, a qualitative data analysis software package. Using Framework analysis, we developed key themes through a systematic process that involved reading, rereading, coding and summarising the transcripts and and subsequent in-depth analysis of the dataset.[6] Final themes were discussed in the research group (HW, JB, SD and TR) and further analysed in relation to the existing literature.

## RESULTS

Fifty-two patients were recruited, 25 at one clinic and 27 at the other. The sample included 41 men and 11 women; 37 men acquired HIV through sex with other men (MSM), while the rest through heterosexual contact (n=14) or injection drug use (n=1). There were 11 in generation 1, 14 generation 2, 17 generation 3 and 10 generation 4. The characteristics of the study participants alongside those of the clinic population for 2014 are shown in table 1.

The generation samples differed somewhat by gender and acquisition: the women were concentrated in generations 1 and 2 (n=6 and 4, respectively), and MSM in generations 3 and 4 (n=16 and 8, respectively).

We have used pseudonyms for each of the following quotes from participants.

### 'Becoming positive': the impact of HIV diagnosis

The experience of receiving an HIV diagnosis was similar across the generations. First reactions were generally of shock and fear of death, irrespective of generation. Alan, diagnosed in 1991, knew nothing about HIV and had not tested before. He remembers vividly the time he received his result:

> I could hear myself saying 'I'm going to die'. Not verbally but in my mind, 'I'm going to die, I'm going to die'. (gen 1, MSM)

Sylvia, diagnosed 2001, was 'totally devastated':

> … I didn't see myself going back and doing my Master's degree for what reason am I going back to do that if I have maybe five years to live. (gen 2, woman)

Roger, diagnosed more than 20 years later, reported several previous tests and considered himself well informed. However, his principal concern on receiving a positive diagnosis was also about life expectancy:

> But even I was not certain. Certainty is the wrong word. I was under the illusion that my expiry date was stamped on me now. (gen 4, MSM)

Fear of a positive result was a key factor in delayed diagnosis for several MSM in all generations, who reported concerns about the impact of HIV on their lives. They were aware of their risk and described feeling relieved at diagnosis as HIV had been 'hanging over them' for years;

**Table 1** Study sample characteristics compared with the clinic cohorts

| | Clinic A | | | | Clinic B | | | |
| --- | --- | --- | --- | --- | --- | --- | --- | --- |
| | Study sample | | 2014 Cohort | | Study sample | | 2014 Cohort | |
| | n | % | n | % | n | % | n | % |
| **Gender** | | | | | | | | |
| Male | 18 | 72.0 | 2459 | 77.5 | 23 | 85.2 | 7743 | 90.3 |
| Female | 7 | 28.0 | 715 | 22.5 | 4 | 14.8 | 830 | 9.7 |
| **Age (years)** | | | | | | | | |
| 18–24 | 0 | 0.0 | 124 | 3.9 | 0 | 0.0 | 186 | 2.2 |
| 25–34 | 4 | 16.0 | 465 | 14.7 | 3 | 11.1 | 1729 | 20.2 |
| 35–49 | 16 | 64.0 | 1502 | 47.3 | 14 | 51.9 | 4164 | 48.6 |
| 50+ | 5 | 20.0 | 1083 | 34.1 | 10 | 37.0 | 2494 | 29.1 |
| **Ethnicity** | | | | | | | | |
| White | 14 | 56.0 | 1541 | 48.6 | 22 | 81.5 | 6401 | 74.7 |
| Black African | 5 | 20.0 | 720 | 22.7 | 5 | 18.5 | 778 | 9.1 |
| Black Caribbean | 0 | 0.0 | 114 | 3.6 | 0 | 0.0 | 201 | 2.3 |
| Other/mixed | 6 | 24.0 | 746 | 23.5 | 0 | 0.0 | 1132 | 13.2 |
| Not reported | 0 | 0.0 | 53 | 1.7 | 0 | 0.0 | 61 | 0.7 |
| **Exposure route** | | | | | | | | |
| Sex between men | 15 | 60.0 | 1971 | 62.1 | 22 | 81.5 | 6776 | 79.0 |
| Heterosexual contact | 9 | 36.0 | 1037 | 32.7 | 5 | 18.5 | 1209 | 14.1 |
| Injecting drug use | 1 | 4.0 | 52 | 1.6 | 0 | 0.0 | 109 | 1.3 |
| Other | 0 | 0.0 | 112 | 3.5 | 0 | 0.0 | 46 | 0.5 |
| Undetermined | 0 | 0.0 | 2 | 0.1 | 0 | 0.0 | 433 | 5.1 |
| **Year of diagnosis** | | | | | | | | |
| Pre-1997 | 6 | 24.0 | 638 | 20.1 | 5 | 18.5 | 1399 | 16.3 |
| 1997–2005 | 6 | 24.0 | 1234 | 38.9 | 8 | 29.6 | 2580 | 30.1 |
| 2006–2012 | 7 | 28.0 | 986 | 31.1 | 10 | 37.0 | 3199 | 37.3 |
| 2013 onwards | 6 | 24.0 | 316 | 10.0 | 4 | 14.8 | 1395 | 16.3 |
| Total | | | 3174 | | | | 8573 | |

the diagnosis confirmed their suspicions. William, who had never tested before, presented with symptoms:

[I had been] burying my head in the sand. I guess I knew I had it but didn't, at the same time, want it confirmed. (gen 2, MSM)

Brian (gen 4, MSM), recently diagnosed, had '*spent on and off probably eight years thinking about it*'. He felt he had '*done all the thinking before*' so, although disappointed, he was also relieved.

Most other participants, particularly heterosexual men and women, were not expecting a positive result and had not requested an HIV test. They were diagnosed either following ongoing symptoms of ill health or having presented for a general sexual health check-up.

None of the women had been diagnosed during pregnancy; most were diagnosed before antenatal screening became routine in the UK (1999). For example, Olivia, diagnosed in 1998, had not been tested in pregnancy. Her 6-month-old baby became sick, and both baby and

husband were then diagnosed with HIV but she did not believe she had HIV and delayed testing for several weeks.

Me I don't have HIV because I never went with other men. (gen 2, woman)

The response and level of support offered by clinicians at this critical time were important to participants' immediate well-being and influenced what happened next, including continuing engagement in care. Martha remarked:

I remember how lovely [name of clinician] was and I've always said I could never wish for a better person to ever tell me or try to guide me, or to reassure me more than what she did because she was perfect. (gen 4, woman)

While most experiences were positive, there were some exceptions. Paul (gen 3, MSM) had regularly tested negative but continued to take risks. Testing positive in 2010 was totally unexpected, leaving him '*numb with shock*', and

he did not feel that he was supported appropriately. The clinician who gave him the diagnosis seemed '[to be on] *autopilot because he had seen people like me before*' and was '*working to his own agenda*'. Despite Paul's obvious distress, the clinician asked him to ring potential contacts during the consultation. Furthermore, when the clinician said: '*Oh this can be managed, don't worry*', Paul interpreted this to mean 'managed to his death'. After 2 weeks of acute anxiety, Paul contacted a friend who was able to reassure him about treatment and the care pathway. Similar experiences led other participants to feel vulnerable, isolated and slow to accept their diagnosis.

### 'Becoming an HIV patient': developing a relationship with clinic and clinician

Once diagnosed, participants described a sense of reassurance about being in the 'best hands', managed by experts in HIV medicine. The majority across the generations described strong relationships with their clinicians and valued seeing the same person each visit. It felt '*like a partnership*' with '*someone you can tell anything*', who knew them and their entire history, ensuring that care went beyond just the clinical management of HIV: '*We seriously talk about how I am not just what my CD4 count is*'.

However, some had not developed a trusting relationship. Marty (gen 3, MSM), for example, diagnosed HIV in 2012, was not eligible for treatment under guidelines at that point. He described anxiety about this lack of treatment, feeling that it adversely affected pre-existing mental health problems that were not addressed by his clinicians. He attended two different clinics and was on the verge of dropping out of care when, as he recounted: '*I basically rescued myself*'. His friend recommended a clinician: '*She got me just like that thank god, thank god…I finally found and she was willing to fight my case*'.

Another participant, Peter, diagnosed in 2009, reported changing HIV clinics within 3 months of diagnosis. He recalled a series of mistakes, miscommunication and a 'dehumanising' clinic environment. Losing trust in clinicians and the service, he finally gained confidence from attending a support group and moved his care:

> I remember that I said, it was like falling off a building… I'm slowly falling backwards as the virus increases. It felt like they were holding a blanket at the bottom to catch me but it felt like they were holding it in the wrong place. I was being asked to trust. (gen 3, MSM)

All participants valued continuity of care, although two recently diagnosed felt that it was not always necessary to see the same clinician. However, continuity was affected by what some described as the very busy clinics limiting the time for consultations and impeding communication.

All participants were in care at the time of the interview but some described having stayed away in the past. Two of six women diagnosed before 1996 had dropped out of care for several years. Given the lack of effective treatment, they had found clinic visits depressing and

preferred to keep away until they became sick. Marie (gen 1, woman) explained, '*I didn't want a life where I just would go to tests and I am scared and they had nothing to offer*'. Alison (gen 1, woman) described the '*terrible situation*' at the clinic, where she saw young gay men, couples, where one would be fit and the other '*in a wheelchair, a skeleton*'. She felt sorry for the doctors, '*there were all these poor young doctors with nothing to offer and seeing these very ill people*'.

Seven of all those diagnosed since ARTs became available described occasional or multiple lapses in attendance; these were generally explained by issues external to the clinic such as recreational drug use, household disruption, mental health problems or competing comorbidities. Re-engagement with care was easier when their clinician actively reached out; for example, some consultants had telephoned patients when they missed appointments and one woman (gen 1) described interventions of this kind as being '*like my family*'. Even though not all patients were contacted when they did not attend services, and one expressed surprise that no one had tried, all found their way back into care.

### 'Becoming medicalised': starting treatment

Almost all (48) of the participants were on treatment; one had stopped medication because of drug interactions and three chose to remain off treatment. Despite the simplification of regimens, participants in all four generations found the decision to start treatment a significant life event.

For earlier generations, the decision had been complex because of toxicity related to ARTs. Some refused treatment contrary to medical advice. Alison, described earlier, recalled, '*In the waiting rooms people said, "don't take it, it will kill you", so I refused that (AZT)*'. Others felt well without treatment like Marie, who was diagnosed in 1986 but only started ARTs 25 years after diagnosis when her CD4 count crashed. She felt she had no choice, '*I fought it all this time on my own, and then finally I had to give in and take a pill. That was kind of depressing*'.

Those diagnosed more recently found it easier to decide to begin treatment, but it was still a significant moment. Tim, diagnosed in 2012, was aware of the latest research and asked to start treatment immediately, even for him, the 'treatment appointment' was a sobering experience:

> [It was] the only time there was a tear. I just thought, God this is a new chapter now. This is a new chapter in my life. I am going to have to take this pill for the rest of my life. (gen 3, MSM)

Brian, diagnosed in 2013, sums up some of the issues that participants said they had considered when deciding to start medication:

> Well the impact it would have on my life, the damage it would do to my body. Would I cope with the medication? Would I be able to continue working? Because so many people have side effects initially and it takes them a long time to get over. I had a lot of

responsibility at work and I couldn't actually manage responsibility well enough once on medication. Would I be able to take the medication on time? Would life's pressures allow me to do what I needed to do? And so on. (gen 4, MSM)

The three participants not on treatment described feeling healthy and wished to remain drug free for as long as possible.

For many participants, HIV occupied only a small part of their lives, but daily medications proved to be a constant reminder of their status even if only for a moment each day. Martin, diagnosed in 2014, described his relationship with his medication:

It's strange sometimes because you look at this pill and you think between you and this little pill lies – it's keeping you alive. And I have never had a pill to take like that before. So, it's very strange. It's my friend and foe at the same time. (gen 4, MSM)

## DISCUSSION
We have found that patients' experiences of, and engagement with, care are influenced by the point at which they were diagnosed, with the earliest cohorts facing substantial challenges. However, we have also found that being diagnosed with HIV and starting treatment continue to be significant life-altering events even in the era of effective and simple treatments. This study brings new insights that are important when considering how future services should be provided.

The revolution in HIV treatment over three decades means it can now be described as a chronic, manageable condition.[7 8] In our study, all our patients were virally suppressed having passed through all the stages of the continuum and arrived on the other side. However, patients' recall of their experience of navigating this journey revealed a range of quite complex issues faced by them at different points, reminding us that many challenges remain in the successful provision of HIV care. We hypothesised that the revolution in treatment would have an impact on the experience of the different diagnostic generations moving through the care continuum, with those diagnosed more recently having a smoother journey. We identified some important differences; the visible illness and deaths from AIDS made it harder for earlier generations to engage with or remain in care and some dropped out, returning only when ill. Decisions about treatment were particularly difficult for the early generations, for whom the association of treatment with severe side effects remained strong until more recently and who expressed pride in, and determination to, remaining well without treatment.

However, our primary finding was a striking similarity across participants' accounts of key stages of the HIV care continuum: diagnosis was a major, traumatic life event for almost everyone, and anticipation of an HIV-positive status

affected the timing of testing and diagnosis. Engagement with care was facilitated by a responsive, flexible approach on the part of services and clinicians, starting with the way the positive diagnosis was handled. Finally, initiating treatment was a major life decision even when recommended by protocol and considered straightforward by clinicians.

Despite the drive to normalise HIV testing through simplified sampling, reduced pretest discussion and expanded test settings,[9–11] receiving an HIV diagnosis remained a significant shock for most participants irrespective of generation, sexuality or gender, as suggested elsewhere.[12–15] Moreover, we found that none of the heterosexual participants were expecting a positive result, a finding that also applied to some of the MSM participants. Resonating with other studies,[13 16 17] we found that fear of imminent death and experience of profound distress did not change, despite the availability of effective and less toxic treatments. This fear, often coupled with a fear of social exclusion and rejection, led some MSM participants who suspected they were positive to delay testing.[18] Bury[19] usefully describes this experience of illness and especially chronic illness as 'biographical disruption'. When everyday life and its meanings are turned upside down, relationships and social networks are disrupted, and plans for the future have to be re-examined.[19] Participants described this type of disruption at diagnosis, whether HIV was considered an acute infection or a chronic condition.

A clinician's approach to a patient with a positive result is considered critical to patients' experiences and may be more important than other aspects of the testing process.[12 15 17 20 21] We found that negative experiences at this critical point affected immediate wellbeing and further contact with services. The impact of those initial encounters, both good and bad, left participants with lasting impressions throughout their journey, thus demonstrating the importance of establishing trust between clinician and patient as a firm foundation for good retention in care.[22–24] Women, a minority in the clinics and, in our sample mainly from the earlier generations, faced particular challenges in engaging in care. This made the establishment of a trusting relationship with their clinician all the more important to managing their quality of life with HIV. Moreover, the importance of that relationship for participants was underlined by their willingness to change their treatment centre until they found what they perceived as a good clinician/patient relationship. The UK policy of open access to any clinic through self-referral may be another explanation for high levels of retention in care.

The prospect of treatment for life, to sustain life, was a major life event. Currently, the moves towards starting treatment at diagnosis, the test-and-treat model, is based on the confidence of biomedicine in HIV management but may be at odds with patient concerns.[25] The British HIV Association interim guidelines 2016 recommend starting treatment on all diagnosed with HIV regardless of CD4 count and continue to recognise that

social, psychological, cultural and economic factors can adversely affect adherence and treatment outcomes.[26] Starting medication on the same day as, or soon after, diagnosis when individuals may be distressed by the positive result could preclude a meaningful discussion of the patient's 'readiness to start'. Persson et al's[27] 2016 study of patients not on ARTs found similar barriers and concerns: for example, logistics of starting life-long medication, fear of long term side effects and desire to stay drug-free while healthy.

Considering ARTs was another point at which some participants anticipated 'biographical disruption', which deterred them from starting treatment.

The study's strengths are in the large number of participants at two different clinics, the inclusion of a range of people with HIV broadly similar to the cohorts seen at these clinics, and diagnosed across the four generations. The semistructured interview format allowed us to explore factors important to participants rather than asking about predefined concerns. However, our study sample was imbalanced between generations, with larger numbers of women in the earliest two generations and only one of those more recently diagnosed. This means that some of our conclusions about generation may also reflect gendered differences. Our recruitment methods that relied on recruitment from the two clinics meant we were not able to explore the experiences of those who remain outside care. The study is also limited by the single interview format that meant people were having to recall past experiences, some of which were three decades ago. This may have introduced recall bias with subsequent experience colouring earlier reports; moreover, earlier generations had more time to have disengaged and re-engaged with care. The focus of the study on a particular model of care in London limits the generalisability of our findings to other settings.

The evolution of simpler treatments has been accompanied by a reconfiguration of the care pathway. In London, HIV clinics are increasingly narrowly focused on HIV and HIV-specific medications, and clinicians are not authorised to provide more holistic medical care.[28] In practice, this has led to less frequent clinic visits, a shift to virtual 'e-clinics' and greater links with, and reliance on, general practitioners. The impact of this on the care continuum is unclear. Continuity of clinician, the atmosphere in the clinic and good communication are recognised to be key issues for patients.[29 30]

There is increasing recognition that viral suppression is not the final goal for people who are now living longer with HIV.[31–33] Lazarus et al[31] have called for a 'fourth 90' 'providing an explicit target for health- related quality of life'.[31] In December 2017, these concerns were embodied in policy recommendations from the European Parliament calling for an integrated and patient-centred approach to long-term HIV care ensuring that services are meeting this challenge.[34] Our study illustrates that the patient journey is complex, and personalised care should not be lost with streamlining pathways. All our

participants were virally suppressed but that can hide the reality of their reliance on the caregivers to help maintain that stability.

A personal, holistic approach has been a hallmark of HIV care since the beginning of the epidemic. It is important that major advances in biomedical treatment do not undermine the care continuum through a loss of care that meets the complex needs of patients, for whom HIV diagnosis and treatment remain significant challenges requiring supportive and flexible care.

**Acknowledgements** Thanks to participants for sharing their time and stories, to Jane Rowlands and Chris Higgs for conducting some of the interviews and the staff of both clinics for facilitating the study.

**Collaborators** Jane Rowlands, Christopher Higgs undertook some of the interviews at one of the two study sites.

**Contributors** All four authors were involved in the design of the study. JB and TR completed the initial analysis and interpretation of the data and subsequent discussion, and analysis was undertaken by all authors through team discussion. JB completed the first draft of the manuscript. All authors participated in critiquing and revising the draft manuscript and all approved the final draft. All authors accept accountability for the submitted paper.

**Funding** Funded by grants from Imperial NIHR Biomedical Research Centre, P35771, the Imperial College Healthcare Charity, P44223, and supported by the St Stephens AIDS Trust. No pharmaceutical grants were received in the development of this study.

**Competing interests** None declared.

**Patient consent** Detail has been removed from this case description/these case descriptions to ensure anonymity. The editors and reviewers have seen the detailed information available and are satisfied that the information backs up the case the authors are making.

**Ethics approval** Ethical approval was obtained from the National Research Ethics Service (NRES) (reference number 14/WM/0147) in May 2014, and research governance approval was obtained from the local sites.

**Provenance and peer review** Not commissioned; externally peer reviewed.

**Data sharing statement** The interview transcripts are anonymised and stored securely at Imperial College London on a password-protected database on a password-protected computer. The participants were assured of the security of their data, and they would remain confidential.

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
