## [Reviewer comments · BMJ Open]

ARTICLE DETAILS

TITLE (PROVISIONAL)	Patient perspectives on the HIV continuum of care in London: a qualitative study of people diagnosed between 1986 and 2014
AUTHORS	Bruton, Jane; Rai, Tanvi; Day, Sophie; Ward, Helen

VERSION 1 – REVIEW

REVIEWER	Shema Tariq Institute of Global Health, University College London
REVIEW RETURNED	08-Nov-2017

GENERAL COMMENTS	I would like to congratulate the authors on an interesting and well-written paper. I enjoyed reading it. I have some relatively minor suggestions to improve the paper further. I have also added comments to the attached annotated PDF. 1. Could you add more justification for this study in the introduction? In what ways did you hypothesise that the experience of diagnosis would change over time, and why? I think this needs to be brought out for a non-HIV specialist audience.2. Please include details of the professional background of the researchers and gender - this is important when interpreting qualitative results.3. Was your analysis informed by any conceptual/theoretical framework? At the very least, I think reference to relevant theory in the discussion would strengthen the paper? Work on biographical disruption, and Annemarie Mol's "The Logic of Care" both spring to mind as relevant pieces here.4. Please state your analytic approach e.g. grounded theory/thematic analysis etc.5. I would have liked to see more on the differences between men and women in terms of experience of diagnosis - I would expect women to be less prepared for a diagnosis (whereas the MSM participants you cite suspected their status). Also I expect many women to have been diagnosed during pregnancy, which again shapes the experience. Can you bring this out?6. In the discussion you may want to refer to new work on the importance of quality of life in HIV, the "fourth 90", and going beyond viral suppression.7. I would like to see a sentence or two clearly stating what new knowledge this study contributes - it certainly does bring new
--

	insights but I think this needs to be articulated clearly for the reader. -The reviewer also provided a marked copy with additional comments. Please contact the publisher for full details.
--	--

REVIEWER	Melissa Withers University of Southern California, Keck School of Medicine Los Angeles, CA, USA
REVIEW RETURNED	12-Nov-2017

GENERAL COMMENTS	Thank you for the opportunity to review this paper describing the results of a qualitative study looking at HIV-positive patients' perspectives relating to their care. I thought this was a very well-written paper on an interesting study that looked at how care for HIV has evolved over time and how this has influenced patient experiences. I had a few minor comments that the authors might want to address:  -some of the acronyms were not spelled out the first time they were used ("UK", for example). Also, sometimes "ARV" was used and another time "ART" was used. So this should be consistent. -The fact that 52 patients in 4 "generations" were interviewed was a major strength. -I liked the inclusion of the question guide, as this helps the reader understand the topics that were featured. -I felt that it would have been helpful to know more about the clinic from which the sample was recruited. What type of clients do they typically serve? How does it work in terms of seeing a regular clinician, etc.? -The biggest problem I had with this paper is the lack of women in the sample. What was the explanation for so few females? Was it simply that the clinics served more males? With only 11 women (and only 1 of these in generations 3 & 4), I don't think that any real conclusions can be drawn about their experiences. If the objective of this study is to examine perspectives and experiences about care across 4 generations, I think the females should be removed from the analysis for this paper. Furthermore, I noticed that only one quote from a woman was used. Therefore, I don't think it would make a significant impact in terms of the results if women were removed. -I also noticed that women were labeled "female" but then men were labeled "MSM." Wouldn't it be more sensitive to label the males as "men" without further saying "MSM"? -Furthermore, it appears that the participants were labeled with their codes (A25, A6, B16, etc). Does this really contribute anything in terms of this paper? I would suggest just deleting this. Or, changing it from the code to something more useful, such as age. -I found a few typos (especially with extra commas, lack of spaces, etc) so I would suggest a thorough review. -In terms of policy implications, I felt the finding that many women viewed the clinics as depressing and couldn't relate to the other patients was especially important. I also felt that some men returned to the clinic after a lapse in treatment when encouraged by a clinician was also very useful. I felt these warranted further discussion in the discussion section.
--

REVIEWER	Samanta Tresha Lalla-Edward Wits Reproductive Health and HIV Institute
-----------------	---

	University of Witwatersrand, Johannesburg, South Africa
REVIEW RETURNED	21-Nov-2017

GENERAL COMMENTS	Manuscript ID bmjopen-2017-020208 "Patient perspectives on the HIV continuum of care in London: a qualitative study of people diagnosed between 1986 and 2014" Bruton J et al COMMENTS TO AUTHORS TECHNICAL ASPECTS Title:  Consider rewording the title – see my comments later on about perspective. Abstract:  Requires revision based on the manuscript revision. Objectives: What you have listed here is more of a background. I suggest you change the heading and include as a last sentence what the objective of this paper is. Methods: two large London HIV clinics Conclusion: A complex sentence where your point is difficult to understand. The sentence in your key statements is nicer, to the point and easier to understand. Introduction:  Pg4 L8 - be diagnosed. Pg4 L26 - perhaps highlight instead of illuminate (sounds like a light shining through). Pg4 L30 – I do not think that perspectives is the correct word. To me perspectives is more for someone on the outside explaining how they see and understand things. In this paper you are describing the insider view/lived experience. In essence you are exploring the experiences of HIV positive clients going through the HIV continuum of care and not a narrative - for instance my view as an HIV negative person of what I think is going on in the care continuum. If you choose to make this change – please edit throughout. Pg4 L31 - perhaps consider explaining this less quantitatively. Hypotheses are not really for qualitative research. The same with using terminology like associated with. There are a few occurrences in the manuscript where this will have to be changed. Methods:
---

- Pg4 L37 – please include a brief description of these two clinics and reasons why they were chosen.
- Pg5 L8 – my understanding is that the FGD was a formative one. Was any of the data collected in the FGD included in these results? Explicitly explain the difference in the formative and actual data collection. Also – did you pilot the questionnaire? If so can you please include a sentence on this and whether the pilot data was included in the analysis.

Results:

- **Table 1:** I understand why you have included this – however you could possibly collapse the Black Caribbean, Other and Not reported into Other.

Discussion

- Pg10 L20 – 42 Consider combining and re-ordering to improve coherence.
- Pg11 – maybe write out UK and BHIVA in full (unless not required by the journal)
- Pg11 L42 – *diverse*. Is your group as diverse as it is representative of the clinics that you recruited them from? Based on table 1 your distribution is similar to the clinic cohorts. It is not diverse from a population perspective (unless there are other characteristics collected that have not been included in this paper/reported elsewhere).
- Are you able to comment on the generalizability of your findings to other settings (local / global)? For instance your findings show that you have a very active health seeking HIV positive population. It is possible that the changes in HIV care in London have no influence on their health seeking and they would continue to access care even if they were treated badly. This is not the case in other settings – this is something that you need to highlight and one of the reasons that your findings may not be generalizable.
- Did you get any data on recommendations for improving the health system for PLHIV?

Overall:

- Overall this is a well written paper about an important, often neglected, aspect of HIV care/service delivery.
- This is a qualitative piece of work – remember that sometimes you need to think and write like a social scientist – not an epidemiologist. If you don't – the richness of the qualitative data is lost.
- Something to think about: London has a controlled HIV epidemic compare to African settings.
 - What are the implications (if any) of your findings to settings like these which 1) have high numbers of

	undiagnosed and diagnosed HIV; 2) high lost to follow up; 3) low viral suppression ADMIN/EDITING  • Title page: Spelling errors with the author affiliations (may need to be changed in the profile – not the paper) • Throughout the manuscript there are complex sentences. Please revisit these together with the punctuation and edit to improve the ease of reading. • Double check throughout – in some instances numbers less than ten are written out in words and in others they appear as figures.
--	--

VERSION 1 – AUTHOR RESPONSE

□Reviewer: 1

Reviewer Name: Shema Tariq

Institution and Country: Institute of Global Health, University College London

1. Could you add more justification for this study in the introduction? In what ways did you hypothesise that the experience of diagnosis would change over time, and why? I think this needs to be brought out for a non-HIV specialist audience.

- We have added further information in the second paragraph, 4th sentence to expand on this, which now reads “However it is unclear whether this change is reflected in patients’ own experiences of passing through each of the stages of care. For example, has the moment of diagnosis become any less traumatic, and have decisions about starting treatment become simpler for patients? Analysis of patient narratives, historically and currently, may help to highlight significant factors for patients in the care continuum.”

2. Please include details of the professional background of the researchers and gender - this is important when interpreting qualitative results.

- The professional backgrounds are included on the title page. Have added the following to methods para 2: “The interviews were semi-structured and carried out by one male and three female researchers (JB, TR, CH, JR), three of whom had clinical backgrounds.”

3. Was your analysis informed by any conceptual/theoretical framework? At the very least, I think reference to relevant theory in the discussion would strengthen the paper? Work on biographical disruption, and Annemarie Mol's "The Logic of Care" both spring to mind as relevant pieces here.

- We did not use an a priori theoretical framework, but adopted an open approach to data collection and initial analysis. However we agree with your suggestion regarding the relevance of notions of biographical disruption which we have addressed in the discussion, with the following additions on diagnosis (paragraph 3) “Bury 1982 usefully describes this experience of illness and especially chronic illness as “biographical disruption”. When everyday life and its meanings are turned upside down, relationships and social networks are disrupted and plans for the future have to be re-examined (Bury M. Chronic illness as biographical disruption. *Sociology of health & illness*. 1982;4(2):167-82.). Participants described this type of disruption at diagnosis whether HIV was considered an acute infection or a chronic condition.” And on starting treatment (paragraph 5) “Considering ARTs was another point at which some participants’ anticipated ‘biographical disruption’ which deterred them from starting treatment.”

4. Please state your analytic approach e.g. grounded theory/thematic analysis etc.

- We used the framework analytical approach and have added this to the methods (para 3) with reference (Gale NK, Heath G, Cameron E, Rashid S, Redwood S. Using the framework method for the analysis of qualitative data in multi-disciplinary health research. BMC Medical Research Methodology. 2013;13(1):117)

5. I would have liked to see more on the differences between men and women in terms of experience of diagnosis - I would expect women to be less prepared for a diagnosis (whereas the MSM participants you cite suspected their status). Also I expect many women to have been diagnosed during pregnancy, which again shapes the experience. Can you bring this out?

- We agree that there are likely to be many differences in experience based on gender, sexual orientation, ethnicity and class as well as the time when people were diagnosed. In this analysis we are focusing specifically on these generations rather than other social determinants as we consider that these are novel findings. However, we have now added some more examples to highlight women's experiences; interestingly most of our women participants were not diagnosed through pregnancy which may reflect the fact that they were mostly diagnosed in the early part of the epidemic (which we note as a limitation of our study). We have added the following:

- Sylvia, diagnosed 2001, was "totally devastated":

"... I didn't see myself going back and doing my Master's degree for what reason am I going back to do that if I have maybe five years to live" (gen 2, woman)

And

- Most other participants, particularly heterosexual men and women, were not expecting a positive result and had not requested an HIV test. They were diagnosed either following ongoing symptoms of ill health or having presented for a general sexual health check-up. None of the women were diagnosed through routine ante-natal screening. For example, Olivia, diagnosed in 1998, had not been tested in pregnancy. Her six-month old baby became sick, and both baby and husband were then diagnosed with HIV but she did not believe she had HIV and delayed testing for several weeks "Me I don't have HIV because I never went with other men" (gen2, woman)

6. In the discussion you may want to refer to new work on the importance of quality of life in HIV, the "fourth 90", and going beyond viral suppression.

- Thank you for this point re fourth 90. Our findings are important especially in recognition of quality of life. See discussion section addition (references included in text below):
"There is increasing recognition that viral suppression is not the final goal for people who are now living longer with HIV. (Lazarus JV, Safreed-Harmon K, Barton SE, Costagliola D, Dedes N, del Amo Valero J, et al. Beyond viral suppression of HIV – the new quality of life frontier. BMC Medicine. 2016;14(1):94. Highlights from the BHIVA Satellite Symposium, IAS Conference, Paris, France, July 2017: 'Tougher times: adapting to increasing demand with declining resources'. Journal of Virus Eradication. 2017;3(4):250-2. Martel K CJ, Auberbach J. Looking Beyond Viral Suppression: Findings from The Well Project's User Survey on Factors Influencing the Health, Well-being, and Quality of Life of Women Living with HIV [IAS 2017 Poster]. 2017 [Available from: <http://www.thewellproject.org/news-press/looking-beyond-viral-suppression-findings-well-project%E2%80%99s-2016-user-survey-factors>). Lazarus et al (2016) have called for a 'fourth 90' "providing an explicit target for health-related quality of life" (see above). In December 2017 these concerns have been embodied in policy recommendations from the European Parliament calling for an integrated and patient-centred approach to long term HIV care ensuring that services are meeting this challenge. (HIV Outcomes, Beyond Viral Suppression. Recommendations Launched at the European Parliament. 2017 [cited 03/01/2018]. [cited 03/01/2018]. Available from: http://hivoutcomes.eu/wp-content/uploads/2017/11/HIV_Booklet_FINAL-DIGITAL-version.pdf). Our study illustrates that the patient journey is complex and personalised care should not be lost with streamlining pathways"

7. I would like to see a sentence or two clearly stating what new knowledge this study contributes - it certainly does bring new insights but I think this needs to be articulated clearly for the reader.

- We have added the following sentences to the start of the discussion: "We have found that patients' experiences of and engagement with care are influenced by the point at which they were diagnosed, with the earliest cohorts facing substantial challenges. However, we have also found that being diagnosed with HIV and starting treatment continue to be significant life-altering events even in the era of effective and simple treatments. This study brings new insights which are important when considering how future services should be provided."

Reviewer: 2

Reviewer Name: Mellissa Withers

Institution and Country: University of Southern California, Keck School of Medicine, Los Angeles, CA, USA

I had a few minor comments that the authors might want to address:

1. some of the acronyms were not spelled out the first time they were used ("UK", for example). Also, sometimes "ARV" was used and another time "ART" was used. So this should be consistent.

- We have spelt out all acronyms in full and used anti-retroviral therapy (ART) throughout.

2. The fact that 52 patients in 4 "generations" were interviewed was a major strength.

- Thank you

3. I liked the inclusion of the question guide, as this helps the reader understand the topics that were featured.

- Thank you

4. I felt that it would have been helpful to know more about the clinic from which the sample was recruited. What type of clients do they typically serve? How does it work in terms of seeing a regular clinician, etc.?

- We have added a little more detail on the clinic and HIV system in the UK at the start of the methods section

"We undertook a qualitative study of people attending two public HIV clinics in London that have provided care since the start of the epidemic; they were also chosen for their large size and diversity, in terms of demographics. They are both specialist HIV clinics linked to sexual health (genitourinary medicine) services. Care is provided by physician-led multidisciplinary teams where patients have a named consultant. HIV care in the UK is free and open access, allowing patients to register at their clinic of choice. "

- Table 1 shows demographics of patients from both clinics.

5. The biggest problem I had with this paper is the lack of women in the sample. What was the explanation for so few females? Was it simply that the clinics served more males? With only 11 women (and only 1 of these in generations 3 & 4), I don't think that any real conclusions can be drawn about their experiences. If the objective of this study is to examine perspectives and experiences about care across 4 generations, I think the females should be removed from the analysis for this paper. Furthermore, I noticed that only one quote from a woman was used. Therefore, I don't think it would make a significant impact in terms of the results if women were removed.

- We agree that there are not enough women although the numbers roughly correspond to the clinic cohorts see table 1. We acknowledge this issue in the limitations section. We had difficulty recruiting women and speculated that this was linked to family responsibilities and lack of time and issues re secrecy. Just one point of correction there is more than one woman quoted in the results. As acknowledged by the reviewer in point 9 (see below) we agree the findings were important such as women finding the clinics depressing and feel it is right to include women in the results to ensure greater representation of the varied experiences of PLWH.

6. I also noticed that women were labelled "female" but then men were labelled "MSM." Wouldn't it be more sensitive to label the males as "men" without further saying "MSM"?

- Thank you for pointing this out. We have now changed "female" to "woman" where appropriate, but continue to use MSM to distinguish from heterosexual men.

7. Furthermore, it appears that the participants were labeled with their codes (A25, A6, B16, etc). Does this really contribute anything in terms of this paper? I would suggest just deleting this. Or, changing it from the code to something more useful, such as age.

- Agreed – deleted codes, pseudonyms used to distinguish different participants

8. I found a few typos (especially with extra commas, lack of spaces, etc) so I would suggest a thorough review.

- Thorough review of draft completed

9. In terms of policy implications, I felt the finding that many women viewed the clinics as depressing and couldn't relate to the other patients was especially important. I also felt that some men returned to the clinic after a lapse in treatment when encouraged by a clinician was also very useful. I felt these warranted further discussion in the discussion section.

- We agree that the issues for women warrant further discussion although as we explain in response to reviewer 1 point 5 gender was not a particular analytic focus. We have added the following to the discussion:

Women, a minority in the clinics and, in our sample mainly from the earlier generations, faced particular challenges engaging in care. This made the establishment of a trusting relationship with their clinician all the more important to managing their quality of life with HIV.

Reviewer: 3

Reviewer Name: Samanta Tresha Lalla-Edward

Institution and Country: Wits Reproductive Health and HIV Institute, University of Witwatersrand, Johannesburg, South Africa

TECHNICAL ASPECTS

Title:

1. Consider rewording the title – see my comments later on about perspective.

- See response to point 8 below

Abstract:

2. Requires revision based on the manuscript revision.

- This has been revised in line with the changes in response to all three reviewers.

3. Objectives: What you have listed here is more of a background. I suggest you change the heading and

include as a last sentence what the objective of this paper is.

- Agreed see revised draft

4. Methods: two large London HIV clinics

- Agreed see revised draft

5. Conclusion: A complex sentence where your point is difficult to understand. The sentence in your key

statements is nicer, to the point and easier to understand.

- Agreed see revised draft

Introduction:

6. Pg4 L8 - be diagnosed.

- Agreed see revised draft

7. Pg4 L26 - perhaps highlight instead of illuminate (sounds like a light shining through).

- Agreed see revised draft

8. Pg4 L30 – I do not think that perspectives is the correct word. To me perspectives is more for someone

on the outside explaining how they see and understand things. In this paper you are describing the insider view/lived experience. In essence you are exploring the experiences of HIV positive clients going

through the HIV continuum of care and not a narrative - for instance my view as an HIV negative person of what I think is going on in the care continuum. If you choose to make this change – please edit throughout.

- We have considered these comments but remain convinced that perspective is appropriate for this paper. The definition of perspective is a way of thinking especially influenced by beliefs and experiences which we think captures the concept of patients' having a view of the treatment continuum based on their own experiences.

9. □□Pg4 L31 - perhaps consider explaining this less quantitatively. Hypotheses are not really for qualitative research. The same with using terminology like associated with. There are a few occurrences in the manuscript where this will have to be changed.

- We are aware that there is a debate regarding the use of hypotheses in qualitative research and agree with those that suggest it is legitimate to have 'assumptions' about the field of interest as the study is not using grounded theory.

Methods:

10. □□Pg4 L37 – please include a brief description of these two clinics and reasons why they were chosen.

- Agreed see revised draft in response to reviewer 2 point 4

11. Pg5 L8 – my understanding is that the FGD was a formative one. Was any of the data collected in the

FGD included in these results? Explicitly explain the difference in the formative and actual data collection. Also – did you pilot the questionnaire? If so can you please include a sentence on this and whether the pilot data was included in the analysis.

- The FGD was formative in the design of the protocol. None of the FGD data collected was in the results. The topic guide was not piloted but informed by the FGD. We have deleted reference to focus group from methods para 2.

Results:

12. □□Table 1: I understand why you have included this – however you could possibly collapse the Black

Caribbean, Other and Not reported into Other.

- These categories are standard in UK data and were provided to use by Public Health England and identifies limitations to our sampling.

Discussion

13. □□Pg10 L20 – 42 Consider combining and re-ordering to improve coherence.

- Thank you, we have made some minor changes to the discussion

14. □□Pg11 – maybe write out UK and BHIVA in full (unless not required by the journal)

- Agreed see revised draft

15. □□Pg11 L42 – diverse. Is your group as diverse as it is representative of the clinics that you recruited them from? Based on table 1 your distribution is similar to the clinic cohorts. It is not diverse from a

population perspective (unless there are other characteristics collected that have not been included in this paper/reported elsewhere).

- Agreed see revised draft “The study’s strengths are in the large number of participants at two different clinics, the inclusion of a range of people with HIV, broadly similar to the cohorts seen at these clinics, and diagnosed across the four generations”

16. □□Are you able to comment on the generalizability of your findings to other settings (local / global)? For

instance your findings show that you have a very active health seeking HIV positive population. It is possible that the changes in HIV care in London have no influence on their health seeking and they would continue to access care even if they were treated badly. This is not the case in other settings – this is something that you need to highlight and one of the reasons that your findings may not be generalizable.

- Thank you. We feel that a broad discussion of this important point is beyond the scope of the paper, but have acknowledged the point by adding another sentence in the limitations: “The focus of

the study on a particular model of care in London limits the generalisability of our findings to other settings.”

17.□□Did you get any data on recommendations for improving the health system for PLHIV?

- Participants did volunteer suggestions about potential improvements but feel that presenting results is beyond the scope and work length of this paper.

Overall:

18.□□Overall this is a well written paper about an important, often neglected, aspect of HIV care/service delivery.

- Thank you

19.□This is a qualitative piece of work – remember that sometimes you need to think and write like a social scientist – not an epidemiologist. If you don’t – the richness of the qualitative data is lost.

- This work has been done by a multidisciplinary team which is reflected in the style of writing which we hope is accessible to a wider audience than social scientists.

20.□□Something to think about: London has a controlled HIV epidemic compare to African settings.

o What are the implications (if any) of your findings to settings like these which 1) have high numbers of undiagnosed and diagnosed HIV; 2) high lost to follow up; 3) low viral suppression

- We accept that this is a specific context but feel that drawing implications for other settings is outside the scope of this paper (see point 16 above)

ADMIN/EDITING

- Thank you for these corrections which have been addressed. The revised paper has been checked throughout.

21.□□Title page: Spelling errors with the author affiliations (may need to be changed in the profile – not the paper)

22.□□Throughout the manuscript there are complex sentences. Please revisit these together with the punctuation and edit to improve the ease of reading.

23.□□Double check throughout – in some instances numbers less than ten are written out in words and in others they appear as figures

VERSION 2 – REVIEW

REVIEWER	Shema Tariq UCL, UK
REVIEW RETURNED	10-Jan-2018

GENERAL COMMENTS	This is a well-written and interesting paper - it has been improved further by the authors' responses to reviewer comments. My only (very minor) comment is that it is important to note that we would not expect many of the female participants to have been diagnosed during pregnancy as they were from earlier generations (pre-dating routine antenatal testing). Perhaps this could be clarified.
--

REVIEWER	Mellissa Withers University of Southern California, USA
REVIEW RETURNED	16-Jan-2018

GENERAL COMMENTS	I feel that the authors have done a very good job of addressing the comments from the three reviewers. The paper has been strengthened and I would recommend it for publication.
--

REVIEWER	Samanta Tresha Lalla-Edward Wits Reproductive Health and HIV Institute, University of Witwatersrand, Johannesburg, South Africa
REVIEW RETURNED	24-Jan-2018

GENERAL COMMENTS	The authors have adequately addressed reviewer comments in this revised version of the manuscript. 2 minor edits in the Methods section: 1) To reflect the evolution of ART (ART has already been defined in the introduction) 2) ...one male (XY) and three female (JB, TR, XX) researchers
---

VERSION 2 – AUTHOR RESPONSE

Response to reviewers' comments

Reviewer 1

My only (very minor) comment is that it is important to note that we would not expect many of the female participants to have been diagnosed during pregnancy as they were from earlier generations (pre-dating routine antenatal testing). Perhaps this could be clarified.

- We have amended the first sentence page 7 paragraph 6:

○ "None of the women had been diagnosed during pregnancy; most were diagnosed before antenatal screening became routine in the UK (1999)"

Reviewer 3

2 minor edits in the Methods section:

To reflect the evolution of ART (ART has already been defined in the introduction)

- We have removed Antiretroviral therapy. See draft methods section page 4 para 1

...one male (XY) and three female (JB, TR, XX) researchers

- We have clarified which reviewers were male and which were female see draft page 5 para 2